# Unravelling Heterogeneity of Amplified Human Amniotic Fluid Stem Cells Sub-Populations

**DOI:** 10.3390/cells10010158

**Published:** 2021-01-15

**Authors:** Francesca Casciaro, Silvia Zia, Mattia Forcato, Manuela Zavatti, Francesca Beretti, Emma Bertucci, Andrea Zattoni, Pierluigi Reschiglian, Francesco Alviano, Laura Bonsi, Matilde Yung Follo, Marco Demaria, Barbara Roda, Tullia Maraldi

**Affiliations:** 1Department of Biomedical, Metabolic and Neural Sciences, University of Modena and Reggio Emilia, 41124 Modena, Italy; francesca.casciaro3@unibo.it (F.C.); manuela.zavatti@unimore.it (M.Z.); francesca.beretti@unimore.it (F.B.); tmaraldi@unimore.it (T.M.); 2Cellular Signalling Laboratory, Department of Biomedical and Neuromotor Sciences, University of Bologna, 40125 Bologna, Italy; matilde.follo@unibo.it; 3European Research Institute for the Biology of Ageing (ERIBA), University Medical Center Groningen (UMCG), University of Groningen, 9713 Groningen, The Netherlands; m.demaria@umcg.nl; 4Stem Sel srl., 40127 Bologna, Italy; silvia.zia@stemsel.it; 5Department of Life Sciences, University of Modena and Reggio Emilia, 41124 Modena, Italy; mattia.forcato@unimore.it; 6Department of Medical and Surgical Sciences for Mothers, Children and Adults, University of Modena and Reggio Emilia, Azienda Ospedaliero Universitaria Policlinico, 41124 Modena, Italy; emma.bertucci@unimore.it; 7Department of Chemistry “G. Ciamician”, University of Bologna, 40125 Bologna, Italy; andrea.zattoni@unibo.it (A.Z.); pierluigi.reschiglian@unibo.it (P.R.); 8Unit of Histology, Embryology and Applied Biology, Department of Experimental, Diagnostic and Specialty Medicine, University of Bologna, 40125 Bologna, Italy; francesco.alviano@unibo.it (F.A.); laura.bonsi@unibo.it (L.B.)

**Keywords:** sorting, amniotic fluid stem cells, transcriptome, stemness

## Abstract

Human amniotic fluid stem cells (hAFSCs) are broadly multipotent immature progenitor cells with high self-renewal and no tumorigenic properties. These cells, even amplified, present very variable morphology, density, intracellular composition and stemness potential, and this heterogeneity can hinder their characterization and potential use in regenerative medicine. Celector^®^ (Stem Sel ltd.) is a new technology that exploits the Non-Equilibrium Earth Gravity Assisted Field Flow Fractionation principles to characterize and label-free sort stem cells based on their solely physical characteristics without any manipulation. Viable cells are collected and used for further studies or direct applications. In order to understand the intrapopulation heterogeneity, various fractions of hAFSCs were isolated using the Celector^®^ profile and live imaging feature. The gene expression profile of each fraction was analysed using whole-transcriptome sequencing (RNAseq). Gene Set Enrichment Analysis identified significant differential expression in pathways related to Stemness, DNA repair, E2F targets, G2M checkpoint, hypoxia, EM transition, mTORC1 signalling, Unfold Protein Response and p53 signalling. These differences were validated by RT-PCR, immunofluorescence and differentiation assays. Interestingly, the different fractions showed distinct and unique stemness properties. These results suggest the existence of deep intra-population differences that can influence the stemness profile of hAFSCs. This study represents a proof-of-concept of the importance of selecting certain cellular fractions with the highest potential to use in regenerative medicine.

## 1. Introduction

Mesenchymal stem cells (MSCs) have the ability to support other cell types during tissue regeneration [1], to secrete various growth factors and cytokines with pro-regenerative properties [2] and to modulate immune responses [3]. For this reason, MSCs therapy represents a potential intervention for a variety of dysfunctions and disorders. A major limitation for their use is a deep inter- and intra- population heterogeneity, which hinders their characterization and affects a consistent therapeutic response [4]. A comprehensive understanding of this heterogeneity represents an essential step towards the definition of their functional properties [5]. A number of variables have been shown to influence human MSCs quality, with age of donor being arguably the most important and a strong limitation for autologous transplantations [6,7,8,9,10].

MSCs are present in both adult and foetal tissues, and more attention has been recently given to placental and amniotic fluid stem cells. These cells express multiple pluripotent markers [11] and differentiate into mesodermal and non-mesodermal lineages under appropriate differentiation conditions, but unlike embryonic stem cells or induced pluripotent stem cells, they do not form tumours in vivo and do not form chimeras when injected into blastocysts [12].

Amniotic fluid (AF) has been used for decades as a diagnostic tool to identify chromosomal aberrations or mutations in the foetus. Importantly, AF contains stem cells from embryonic and extra-embryonic origins that can be isolated and stored for future cell therapy. The majority of Human Amniotic Fluid Stem cells (hAFSCs) shares a multipotent mesenchymal phenotype with higher proliferative potential and a wider differentiation capacity compared to adult MSCs [12]. However, a comprehensive characterization of hAFSCs-associated molecular signatures and of conditions required to maintain and enhance their regenerative potential is still lacking [13].

A major hurdle for this characterization is due to the high hAFSCs heterogeneity, in part due to their derivation from different organs such as urinary and gastrointestinal system, respiratory tract, skin and amniotic membrane [14,15]. Purification of specific populations using stem cells markers, such as the stem cell factor c-kit, can help isolation of cells with the highest multipotency, but still many markers are co-expressed by cells with different origin and properties [16].

Heterogeneity is a biological hallmark of most stem cell populations and a limitation for the development of standard procedures. The heterogeneous behaviour becomes even more evident during ex vivo expansion, a necessary step for generating enough cells for clinical use, and has unpredictable outcomes in terms of cellular aging or senescence [17]. In order to define whether different sub-populations exist within the same hAFSCs preparation, we use here a fractionation protocol based on the Non-Equilibrium, Earth Gravity Assisted Dynamic Fractionation (NEEGA-DF) method, a label-free, flow-assisted method of purifying, distinguishing and sorting MSCs based only on cell physical characteristics [18,19].

Using this system, we isolate and profile different fractions to clarify the level of biological heterogeneity.

## 2. Materials and Methods

### 2.1. Amniotic Fluid Collection

The hAFSCs were obtained from 10 amniotic fluids collected from pregnant women (mean age 37.1 ± SD 4.0), between the 16th and 17th week of gestation, who underwent amniocentesis for maternal request (not for foetal anomalies) at the Unit of Obstetrics and Gynaecology, IRCCS-ASMN of Reggio Emilia and at the Policlinico Hospital of Modena (Italy). The amniocenteses were performed under continuous ultrasound guidance, in a sterile field, with 23-Gauge needles. The risks related to the procedure and the purpose of the study were explained to all patients before the invasive procedure, and the ob-gyn specialist collected a signed consent before starting the exam (protocol 2015/0004362 of 24 February 2015 and protocol 360/2017 dated 15 December 2017 approved by Area Vasta Emilia Nord).

For this study, supernumerary (unused) flasks of AF cells cultured in the Laboratory of Genetics of TEST Lab for 2 weeks were used.

### 2.2. Adult Human Tissue Isolation and Cell Culture

hAFSCs were isolated as previously described [20]. Human amniocentesis cultures, namely the supernumerary flasks of AF, were harvested by trypsinization, and subjected to c-kit immunoselection by MACS technology (Miltenyi Biotec, Germany). hAFSCs were subcultured routinely at 1:3 dilution and not allowed to grow beyond the 70% of confluence. hAFSCs were grown in culture medium (αMEM) supplemented with 20% foetal bovine serum (FBS), 2 mM L-glutamine, 100 U/mL penicillin and 100 μg/mL streptomycin (all from EuroClone Spa, Milano, Italy).

### 2.3. Cellular Proliferation

Cells were seeded in a T25 cm^2^ flask at a density of 2000 cells/cm^2^, cultured for 3.5 days then detached, counted and seeded again at 2000 cells/cm^2^. Cultures were performed until passage 8 and population doubling (PD) for each passage was measured applying the formula reported in [21]

### 2.4. hAFSCs Sub-Populations Sorting

hAFSCs sub-populations were sorted using a label-free, flow-assisted fractionation protocol through Celector^®^ technology (Stem Sel Ltd., Bologna, Italy).

The laminar flow of mobile phase elutes in the capillary device and injected cells reach a specific position across the channel thickness during transportation due to the combined action of gravity and opposing lift forces that depend on the sample’s morphological features. Cells acquire well-defined velocities and are therefore eluted at specific times. This soft fractionation mechanism guarantees the preservation of the native physical features of the cells, and viable, intact cells can be collected at the channel outlet.

A camera with a microscopic object is placed at the outlet of the capillary channel, and it is connected to an imaging software that records live images of eluting cells and plots in a fractogram the number of counted cells related to elution time (profile). The fractionation system was firstly decontaminated and subsequently, in order to block unspecific interaction sites on the plastic walls, a sterile coating solution was flushed at 1 mL/min. The system is then ready to be used after filling it with sterile mobile phase. Stem Sel ltd. (Bologna, Italy) provided all solutions.

Cells were resuspended in an appropriate volume of PBS to obtain a concentration of 300,000 cells per in 100 µL, and then this volume was introduced into the system and analysed at a flow rate of 1 mL/min.

Based on sample profile, hAFSCs populations were divided in fractions and cells were collected from repeated runs, pooled and used for experiments.

### 2.5. Immunofluorescence and Confocal Microscopy

For immunofluorescence analysis, hAFSCs from different fractions were processed, and confocal imaging was performed using a Nikon A1 confocal laser scanning microscope, as previously described [22].

Primary antibodies to detect Oct4, β3-Tubulin and SSEA4 (Cell Signaling, Danvers, MA, USA), Ki-67 (Santa Cruz Biotechnology, Santa Cruz, CA, USA), Runx2 (Abcam, Cambridge, UK), Osteocalcin and GFAP (Millipore, CA, USA), MAP2, nestin, CD29/Integrinβ1 and CD44/HCAM (Santa Cruz Biotechnology), CD271/NGFR (Sigma-Aldrich, St. Louis, MO, USA) were used following datasheet recommended dilutions. Alexa secondary antibodies (Thermo Fisher Scientific, Waltham, MA, USA) were used at 1:200 dilution.

The confocal serial sections were processed with ImageJ software to obtain three-dimensional projections. The image rendering was performed by Adobe Photoshop software.

The cell fluorescence signal was quantified using ImageJ and applying the following formula:

Corrected Total Cell Fluorescence (CTCF) = Integrated Density—(Area of selected cell X Mean fluorescence of background readings).

### 2.6. Differentiation Protocol

Osteogenic differentiation was obtained after 3 weeks of culture in a medium composed by αMEM supplemented with 10% FBS, 2 mM L-glutamine, 100 U/mL penicillin and 100 μg/mL streptomycin (all from EuroClone Spa, Milan, Italy), 100 µM 2P-ascorbic acid, 100 nM dexamethasone and 10 mM β-glycerophosphate (all from Sigma-Aldrich, St. Louis, MO, USA) [23]. The medium was changed twice a week.

Neurogenic differentiation protocol [24]: cells cultures (seed at 60% of confluence) were maintained in neurogenic medium (culture medium 10% foetal bovine serum supplemented with 20 μM retinoic acid in dimethyl sulfoxide) for up to 10 days in an incubator at 37 °C with 5% CO_2_ (all from Sigma-Aldrich, St. Louis, MO, USA).

### 2.7. Alizarin Red S Staining and Mineralization

Fixed monolayer cells were washed with distilled water and then incubated with a 2% of Alizarin Red S solution at pH 4.2 for 10 min at RT. Images of histological samples were obtained with a Zeiss Axiophot microscope (Zeiss AG, Jena, Germany), equipped with a Nikon DS-5Mc CCD color camera.

To quantify the Alizarin Red S staining, stained samples were washed three times with PBS and then 1 mL of 10% cetylpyridinium chloride was added to each well and incubated for 20 min to elute the stain. 100 µL of this eluted stain were added to 96 well plates and read at 485 nm using a spectrophotometer (Appliskan, Thermo Scientific, Vantaa, Finland) [25].

### 2.8. CFU Analysis

In order to evaluate the clonogenic potential, a colony-forming unit (CFU) assay was performed. hAFSCs fractions were plated at 80 and 160 cells/cm^2^ in 12-well tissue culture plates and stored in basal culture medium for 7 days at 37 °C in a 5% CO_2_ humidified atmosphere. The medium was then removed, and the cells were fixed with methanol/acetic acid 3:1 for 5 min, and stained with 0.5% crystal violet in methanol for 30 min at room temperature [26]. Positive CFU was identified as an adherent colony containing at least 50 cells and was visualized using a Nikon TE2000 inverted microscope at 10× magnification. After 10 days, the number of CFU colonies was counted.

### 2.9. Senescence Assay

In order to evaluate the presence of senescent cells in fractions of hAFSCs, cells at passage 6–8 were seeded in 12-well plates and processed using a senescence β-Galactosidase staining kit (Cell Signaling, Danvers, MA, USA), according to the manufacturer’s instructions.

### 2.10. RNA Isolation and Quantification

RNA of collected cells for each fraction was extracted using the ISOLATE II RNA Mini Kit (Bioline Meridian Bioscience, Paris, France) following manufacturer’s instructions.

Briefly, cells were lysed using 350 µL of Lysis Buffer RLY with 3.5 μL β-mercaptoethanol in the presence of guanidinium thiocyanate. After homogenization, ethanol was added to the samples that were processed through a spin column containing a silica membrane to which the RNA binds. During the isolation procedure, genomic DNA contamination was removed by an on-column DNase I digestion at room temperature for 15 min. Any impurities such as salts, metabolites and cellular components were effectively removed by simple washing steps with two different buffers. High-quality purified total RNA was eluted in RNase-free water and quantified at Nanodrop^TM^.

### 2.11. cDNA Retrotranscription and qRT-PCR

500 ng of total RNA were reverse transcribed into cDNA using the kit High Capability cDNA Reverse Transcription (Applied Biosystems, Waltham, MA, USA). The reaction was accomplished in a T100™ Thermal Cycler (Bio-rad, Hercules, CA, USA).

The obtained cDNA samples were diluted at 5 ng/µL and 12 ng of each sample were loaded to perform qRT-PCR using the Universal Probe Library system (Roche, Basel, Switzerland) and SENSIFast Probe kit (Bioline Meridian Bioscience, Paris, France). The reaction was carried out using the LightCycler^®^ 480 Instrument II (Roche, Basel, Switzerland) in a 360 PCR multi-well plate. Reference genes tubulin and actin were used to normalize the expression of other genes’ CT values. The list of used primer is in Table 1.

### 2.12. RNA and DNA Quality Check

Prior library preparation for RNA seq, RNA quality was checked using RNA 6000 Pico kit (Agilent, Santa Clara, CA, USA) following the instructions of the producer and reading the results through a 2100 Bioanalyzer Instrument (Agilent, Santa Clara, CA, USA).

The same procedure was applied to verify both the correct fragmentation of DNA during the library preparation and to quantify the obtained fragments, using in this case the High Sensitivity DNA Kit (Agilent, Santa Clara, CA, USA).

Quantification of DNA fragments was also performed using Qubit™ 1X dsDNA HS Assay Kit (Thermo Fisher Scientific, Vantaa, Finland) at Qubit™ Fluorometer (Thermo Fisher Scientific, Vantaa, Finland), following kit instructions.

### 2.13. Library Preparation and RNAseq

100 ng of total RNA were used to isolate mRNA using NEXTflex™ Poly(A) Beads (Bioo Scientific, Burleson Rd, Austin, TX, USA), as reported in the producer protocol.

Magnetic based separation was used to retain poly(A) mRNA while removing all other transcripts. Beads were subsequently washed and 14 µL of mRNA were eluted, releasing purified poly(A) mRNA.

The obtained mRNA was used to prepare the RNA library using NEXTflex™ Rapid Directional qRNA-Seq™ Kit (Bioo Scientific, Burleson Rd, Austin, TX, USA), following the instructions.

This kit efficiently generates libraries equivalent to conventional directional, strand-specific RNASeq libraries, but with the added feature of Molecular Indexing™.

Briefly, the mRNA was fragmented using a cationic buffer. Fragmented RNA underwent first and second strand synthesis, followed by adenylation, indexed adapter ligation and PCR.

Directionality was retained by adding dUTP during the second strand synthesis step and subsequent cleavage of the uridine-containing strand using Uracil DNA Glycosylase. The strand sequenced is the cDNA strand.

After the quality check and quantification performed with the Bioanalyzer and Qubit, as previously described, samples were pooled together in a unique tube at 2 nM.

At this point, pooled samples were sent to the RNAseq facility and pair-end run Illumina nextseq (high output) sequencing was performed.

Quality control of all samples was performed using the FastQC software v0.11.5. Samples were aligned to the UCSC hg38 version of the human using STAR-2.5.3a aligner, and a count table was directly obtained with Star. Counts for UCSC annotated genes were calculated from the aligned reads using featureCounts function of the Rsubread R package.

Differential analysis was performed using edgeR algorithm. Raw counts were normalized to obtain Counts Per Million mapped reads (CPM) and Reads Per Kilobase per Million mapped reads (RPKM). Only genes with a CPM greater than 1 in at least 1 sample were retained for differential analysis. The heterogeneity of the data was evaluated with a Principal Component Analysis (PCA)-plot of the log-transformed normalized CPM, evaluating whether they clustered with similar samples in the same dataset.

Enrichment analysis was obtained with the software Gene Set Enrichment Analysis (GSEA 3.0) (http://software.broadinstitute.org/gsea/index.jsp), a computational method that determines whether a priori defined set of genes shows statistically significant, concordant differences between two biological states. The gene sets are defined based on prior biological knowledge and here the hallmark, kegg and reactome gene sets collections were used. To analyse stemness signature, the list “ES exp1”, reported elsewhere [27], was used. The results (FDR < 0.25), in which F2, F3 or F4 from all patients were put together and considered as replicates, keeping in mind patient-patient variability, were obtained (both comparing individually each sample and with a paired analysis). Investigation of the biological processes involved in each pathway was performed using the software FunRich 3.1.3 (http://www.funrich.org/).

### 2.14. ELISA Assays

Concentrations of HGF, TGFβ, IDO, IL-6, IL-10 in the media, conditioned for 4 days, of fractionated hAFSCs, were measured by using ELISA assays, according to the manufacturer’s instructions (Boster Biological Technology, Pleasanton, CA, USA; MyBioSource, Peachtree Corners, GA, USA). Depending on the sample, a different cell number was collected in initial and final eluate fractions, since every sample is characterized by a different cell composition. However, cells were seeded at the same density, and hAFSC subfractions were cultured without serum for 4 days. One mL of conditioned medium was derived from 50,000 cells. Similar protein concentration was found in all conditioned medium (CM), analysed by Bradford test. Conditioned media were centrifuged before testing. Samples were run in duplicate. A standard curve was constructed using known concentrations of recombinant human standards.

### 2.15. Mononuclear Cell Separation

Human peripheral blood mononuclear cells (PBMC) were separated from peripheral blood of healthy donors by gradient centrifugation on Ficoll-Hypaque (Lymphoprep, AXIS-SHIELD PoCAs, Oslo, Norway) at room temperature (RT) [28].

The concentration of isolated PBMC was adjusted to 2 × 10^6^ cells/mL in RPMI 1640 (EuroClone Spa, Milan, Italy) including 10% FBS. Twenty hours later, PBMC were washed and used for experiments described below.

### 2.16. Apoptosis Assay of PBMC Exposed to AFSC CM

Co-cultures with the CM derived from initial or final fractions were obtained by adding the 1 mL CM, deriving from the same number of cells, to 1 mL of PBMC suspension (2 × 10^6^ cells/mL) for 4 days.

PBMCs (10^6^ cells) were removed from co-culture and washed with phosphate-buffered saline, and 500 μL of this suspension (0.5 × 10^6^ cells) was washed 3 times with 500 μL of binding buffer (10 mM HEPES, pH 7.5, containing 140 mM NaCl and 2.5 mM CaCl_2_), the specimen was incubated for 15 min with 100 μL of double staining solution (binding buffer containing 1 μL of annexin FITC and 1 μL of propidium iodide (PI); Sigma-Aldrich). Finally, the specimen was washed 5 times with 500 μL of binding buffer, mounted with 100 μL of binding buffer, and visualized under fluorescence microscopy.

### 2.17. SDS PAGE and Western Blot

Whole cell lysates from PBMCs were processed as previously described [28]. Primary antibodies were raised against the following molecules: Actin (Sigma-Aldrich, St. Louis, MO, USA), PARP (Santa Cruz Biotechnology, CA, USA), caspase-7 (Cell Signaling Technology, Lieden, The Netherlands). Secondary antibodies, used at 1:3000 dilution, were all from Thermo Fisher Scientific (Waltham, MA, USA).

### 2.18. Statistical Analysis

In vitro experiments were performed in triplicate. For quantitative comparisons, values were reported as mean ± SE based on triplicate analysis for each sample. To test the significance of observed differences among the study groups, Student’s *t*-test or One-way Anova with Bonferroni post hoc test were applied. A *p* value < 0.05 was considered to be statistically significant. Statistical analysis and plot layout were obtained by using GraphPad Prism^®^ release 6.0 software.

### 2.19. Ethics Approval and Consent to Participate

An informed consent allowing the use of clinical data and biological samples for the specified research purpose (protocol 2015/0004362 of 24 February 2015 and protocol 360/2017 dated 15 December 2017 approved by Area Vasta Emilia Nord) was signed by all infertile couples before treatment and collected by the Unit of Obstetrics and Gynecology, IRCCS-ASMN of Reggio Emilia and by Policlinico Hospital of Modena (Modena, Italy).

## 3. Results

### 3.1. Sorting of hAFSCs by Celector^®^ Technology

c-kit^+^-hAFSCs at passage 5–6 in culture from 10 patients were sorted using the Celector^®^ technology. The graphical output (profile), which is the number of counted cells vs. elution time, showed two types of cell populations: the first one (type 1) showed a Gaussian distribution, with cells eluting between 1 and 4.5 min (Figure 1A), and the second one (type 2) showed a slight longer time interval, with cells eluting from 1 to 5 min (Figure 1B). Sorting was performed based on cell appearance on live camera and profile. When the population is more normally distributed along the elution time, it was divided in three fractions (type 1, Figure 1A). Fraction 1 (F1) (from 1 to 2 min) contained bigger cells and small aggregates, as captured on live images; cells in fraction 2 (F2) (from 2 to 3:30 min) were slightly smaller and cells in fraction 3 (F3) (from 3:30 to 4:30 min) were smaller with a more rigid and defined perimeter (Figure 1A). Type B hAFSCs showed a diverse profile, which is a synonym of a different cell heterogeneity among the population (Figure 1B). These hAFSCs were divided in four fractions, with fraction 1 containing large cell aggregates (F1, from 1 to 2:30 min), followed by bigger cells with jagged perimeter (F2, 2:30 to 3:30 min) and then by cells in F3 and F4 that were smaller and more defined (F3, 3:30 to 4:30 and F4, 4:30 to 5:30 min) (Figure 1B). These data suggest that, even derived from similar gestational age and same selection procedure (c-kit^+^), hAFSCs give rise to heterogeneous sub-populations. In order to identify a possible trend among hAFSCs fractions in all samples, downstream analyses outputs were interpreted clustering the sorted cells in the initial eluate part, the ascending curve, and the final eluate, the descendent part. The initial eluate (F1 + F2) and final eluate (F3 from first profile and F3 + F4 from second profile) are shown in the overlaid profile (Figure 1C).

The comparison of unfractionated hAFSCs proliferation rate, as the population doubling time (PDT), and the type of profile showed an interesting correlation. Cell distribution was calculated as the area under the curve because, as already mentioned, the profile is generated by the number of eluting cells versus time. Slow replicative hAFSCs populations (red circled) were represented by a profile having a higher distribution of cells in the initial eluate part, while fast growing hAFSCs (black circled) showed precisely the opposite, with the majority of cells in the last eluate part (Figure 1D,E).

Initial and final eluate cells showed significant differences in size and morphology both in suspension (Figure 1A,B) and in adhesion (Figure 1F,G), with bigger cells in the initial eluate and smaller ones in the final eluate part.

### 3.2. Specific Characteristics of hAFSCs Fractions

When we compared expression of proliferative markers in the different eluates, we observed that cyclin E2 transcript levels were similarly expressed (Figure 2A), while the expression of Ki67 protein, which reaches its maximum during the M phase [29], was significantly higher in the final eluate (Figure 2B).

As last eluted cells showed different morphological and proliferative profiles, we set to analyse markers of cellular aging.

Enlarged morphology and reduced cell cycle progression are considered hallmarks of cellular senescence. Major regulators of cellular senescence are the cyclin-dependent kinase inhibitors p16INK4A and p21WAF1 [30]. Interestingly, we observed a significant increase of p21 expression in the initial eluate cells (Figure 2C) and higher staining for β-galactosidase activity test (Figure 2D).

Senescent cells undergo metabolism alterations, resulting in an increase of the AMP:ATP and ADP:ATP ratios [30]. The rise of AMP levels causes AMP kinase allosteric activation, which acts as a sensor of the reduced energetic state, further activating catabolic pathways while inhibiting biosynthetic ones, and regulating p53 and other targets [31]. When we measured AMPK transcript level, we observed a lower expression in the final eluate cells, demonstrating an enhanced bioenergetic state of F3/F4 cells (Figure 2E).

Since we have shown that improved proliferative potential and reduced cellular aging is associated with enhanced stemness properties in hAFSCs, we evaluated the expression of the pluripotency markers Oct4 and Nanog. Quantitative PCR analysis revealed increased Oct4 expression, but not Nanog, in the final eluates, as shown by elevated transcript and protein levels (Figure 2F,G). Immunofluorescence staining, shown in Appendix A, revealed that another pluripotent marker expressed during early embryonic development, SSEA4, was upregulated in the final eluted cells (Figure 2G).

A high but variable expression was demonstrated for the mesenchymal markers CD29, CD44 [26] and CD271 [32] in hAFSCs. Interestingly, expression of these three mesenchymal markers was downregulated in the final eluted cells (Figure 2H and Appendix A).

All these data suggested the higher stemness and plasticity of final eluted cells, corroborating the idea that an enlarged and more flattened morphology is typical of cells approaching cellular senescence and loss of plasticity [30].

### 3.3. Transcriptomic Profiles of Different hAFSCs Fractions

Immunofluorescence and gene expression analysis hinted the final eluted cells as the fraction with the higher pluripotent characteristics and proliferation ability, also at late passages.

To further expand the molecular characterization of the different fractions, we performed RNA sequencing (RNAseq) analysis using cellular fractions isolated from different isolated populations.

Principal Component Analysis (PCA) revealed that major variances tended to occur among donors, but differences were identified even between fractions from each population (Figure 3A). Furthermore, to emphasize these differences among fractions, *Principal Component* and *Pearson correlation analyses* were performed on each sample, individually. In particular, the Pearson coefficient was always less than 1, confirming successful isolation of diverse fractions from individual hAFSCs populations (Figure 3B).

Next, we performed a *Gene Set Enrichment Analysis* using datasets from the fractions F2, F3 and F4. The F1 fraction was excluded because of the high content of cell aggregates, a confounding factor tor the analysis. 18 pathways were found to be consistently regulated in F3/F4 cells compared to F2 (Figure 3C), suggesting a similar molecular profile between cells in the F3 and F4 fractions. Most common genes fell into categories of *DNA repair*, *E2F targets*, *G2M checkpoint*, cell cycle progression, and DNA quality check and transcription, which was in support of our initial hypothesis that the proliferative potential of these fractions was significantly improved.

Last eluted cells overexpress genes related to hypoxia and mTOR signalling in support of metabolic alterations.

As protection from the cellular stress, final eluted cells were supported by lower expression of genes associated to the *Unfolded Protein Response (UPR)* [33]. Differences in expression of mesenchymal proteins in the final eluted cells, shown in Figure 2G, were confirmed by a general downregulation of genes involved in extracellular matrix organization (categorized in the *Mesenchymal Epithelial transition pathway).*

While there was a strong overlap in gene expression between F3 and F4 cells, we also observed differences. For example, *p53* expression was downregulated in F4 compared to F2, while the opposite behaviour was observed in F3 cells. However, when we used two different analysis tools, KEGG and Reactome, we observed that the p53 pathway was downregulated both in F3 and F4 cells compared to F2 (Figure 4A). Indeed, when genes composition of this pathway was processed using the three datasets, a huge difference was discovered in the type and number of genes involved. According to the general dataset Hallmark, 200 genes were involved in p53 function as regulators of cell cycle and senescence but to a larger extent in other processes like apoptosis, whose regulation in senescence is still debated [34] mostly due to post translational processes. Only 3 genes of Hallmark dataset were found in common with KEGG and Reactome results, which included a smaller number of genes, 69 and 57, respectively. In particular, Reactome focuses on cell cycle regulation and NF-kB pathway, which has been reported to interact at multiple levels with the UPR [35] (Figure 4B). For these reasons, we believe that Reactome better describes the processes that we are interested in.

Since no stemness signature was available in any of the three datasets previously used, we applied the “ES exp1” list [27], which contains the original gene sets of embryonic stem (ES) cell: 380 genes and all three stem cell master regulators (Oct4, Nanog and Sox2). Importantly, the stemness pathway was upregulated in the final eluate cells (F3 and F4), confirming the quantitative PCR and immunofluorescence results shown in Figure 2F,G.

### 3.4. Clonogenic and Differentiation Ability of hAFSCs Fractions

We then decided to further explore the stemness properties of the various fractions. Using clonogenic assays, we showed that cells from final elute formed more CFU-F, supporting the transcriptome analysis of the whole ES pathway as well as immunofluorescence data on Oct4 and SSEA4 (Figure 5A).

Next, we tested the capacity of the various fractions to differentiate towards osteogenic lineage. hAFSCs from final and initial eluates were exposed for 3 weeks to the appropriate differentiation media and expression of the tissue specific protein Runx2, by immunofluorescence (Figure 5B–D), and alizarin red staining, to reveal the mineralized matrix, were evaluated.

In non-differentiating condition, the initial eluate cells showed a higher formation of mineralized extracellular matrix over cell monolayer, as compared with final eluted cells. Moreover, the expression of Runx2, an early marker of osteogenesis, was higher in initial eluate fraction. These results are consistent with the immunofluorescence data on mesenchymal marker expression and the downregulation of *Mesenchymal Epithelial transition pathway* observed for final eluted samples.

However, upon exposure to differentiation media, final eluate cells took over the initial eluate cells, as shown in the alizarin red staining as well as osteocalcin expression that were higher in the final eluate cells, showing that the final eluted cells, even if less committed to mesenchymal differentiation, preserve a great differentiation plasticity.

Finally, we tested the neurogenic commitment: hAFSCs from final and initial eluates were exposed for 10 days to the appropriate differentiation media and expression of the neuron and astrocyte specific proteins were analyzed by immunofluorescence. All the tested markers (Figure 6), namely nestin, β3-Tubulin, Microtubule Associated Protein 2 (MAP2) and Glial fibrillary acidic protein (GFAP) were more expressed in the final eluate, confirming its better differentiation capability.

### 3.5. Analysis of the Secretome Produced by hAFSC Fractions

In order to quantify the presence of immune-modulating factors in the secretome produced by the two hAFSC sub-populations, ELISA tests were performed (Figure 7A). These analyses demonstrated that the condition medium (CM) of final eluted cells contained higher level of TGFβ and HGF compared to the initial eluted cells, while IDO, IL-10 and IL-6 were statistically similar.

In previous work, we demonstrated that CM of hAFSCs induces apoptosis in PBMCs [36] especially by HGF action, which is known to have a diverse functional effect depending on cell context, and thus both apoptotic and antiapoptotic effects [37]. Immunomodulation properties by condition medium are still not completely understood; here we tried to dissect the effect of secretome of the two hAFSCs sub-populations obtained after the sorting process. We designed the assay as previously tested. In brief, CM were collected after 4 days in culture (deprived of serum). Then, PMBCs were exposed to CM for another 4 days. In Figure 7B, the graph represents the positivity for Annexin V and PI: PBMCs exposed to final eluate cells’ CM showed a higher number of positive cells for the early and late apoptotic markers, indicating the onset of an apoptotic process. The same samples of PBMCs were analysed by western blot for other apoptotic markers (Figure 7C): the cleaved form of PARP and caspase 7 significantly increased in the presence of final eluate CM compared to the initial eluted CM. The increase of these markers confirms the presence of apoptotic death.

## 4. Discussion

Previous reports showed inter-patient heterogeneity in hAFSCs, even when specific sub-populations, such as the CD117 (c-kit)+, were analysed [21,26,38]. Various studies identified a clear cell-to-cell variability within a single MSC population [39]. This variability becomes more evident during in vitro expansion, where sub-populations with different proliferative and morphological properties emerge [40]. Romani et al. [41] characterized two distinct types of stem cells from residual amniotic fluid (AF) from prenatal diagnostic amniocentesis based on morphological characteristics. The two types of cells differed in their morphology and growth kinetics, resulting in slow-growing oval-shaped cells and fast-growing fibroblastic-shaped cells. Both populations expressed pluripotent stem-cell markers but differences in clonogenic potential, expression of KLF4, SSEA-4 and CD117 markers, and proteomic patterns were demonstrated.

These differences influence stemness capability, the differentiation potential and the predisposition to senescence process.

When single cells are analysed using a high-resolution technology based on specific features (morphology, marker expression, cell cycle stage), heterogeneity was observed. It is accepted that populations in culture are the consequence of multiple clones, with the challenge being the definition of each clonal biological property [42]. For this reason, it is necessary to decompose the heterogeneity of cell population using interpretability and comprehensive dimensions. The goal of decomposition is to identify what clones maintain the highest regenerative and differentiative potential.

In this work, we investigated an innovative approach to decompose heterogeneous cell population, with a focus on finding sub-populations with the optimal characteristics for in vitro expansion and regenerative medicine application. We tested a fractionation protocol based on a label free, flow-assisted method to purify, distinguish and sort MSCs from complex samples [18,19].

hAFSCs were divided into several fractions, and each fraction was phenotypically analysed. First, we identified differences among the obtained profiles, which confirmed inter-patient heterogeneity. Second, by analysing two main fractions separated by cellular properties, we defined sub-populations with different biological functions.

We found out that dimension and cell contour of cells in suspension is a predictive tool for stemnsess. Bigger and jagged cells, which were present in the initial eluted fractions, were less staminal as compared to cells smaller in size and with a more defined membrane contour, present in the second eluted fraction. Larger cells showed a mesenchymal phenotype and lower expression of the staminal marker Oct4. Accordingly, these cells showed lower differentiation properties compared to cells in the final elute.

Cell shape has been extensively studied in adherent cells, where the type of surfaces was also proved to profoundly change the physiologic state and induce a commitment towards a specific lineage. When embryonic stem cells were cultured in surfaces with a diameter smaller than the cell diameter, they conserved their original round shape and remained undifferentiated [43]. ESCs are very small, with a higher nuclear/cytoplasmic ratio and a defined perimeter. These characteristics belonged to the final eluted cells, which were proven to have higher proliferative potential, improved replicative lifespan and enhanced expression of embryonic stem cells markers Oct4 and SSEA4. Transcriptomic analysis showed up-regulation of ES pathway in final eluate cells, together with upregulation of the Hypoxia pathway and downregulation of the mTORC1 signalling, suggesting altered metabolic profile and increased mitochondrial biogenesis and activity [44,45].

The profile and the percentage of cells belonging to the initial and final eluate could be a predictive tool of stemness quality. Samples with a predominance of final eluted cells also displayed a better proliferation ability. As demonstrated by analysis of cell cycle related proteins, Ki67 expression was upregulated in the last eluted cells supporting the transcriptome evaluation of E2F targets and G2M checkpoint that were upregulated in these fractions. In parallel, senescence markers, in particular the p53 signalling pathway and the classical p53 target gene, p21, were decreased. Opposite behaviour was seen in samples with a predominance of initial eluted cells that showed an enlarged morphology and slower proliferation. These aspects are considered hallmarks of cellular senescence [30] and were identified in one third of the whole cohort of hAFSCs we tested in the last 5 years [26,36].

Since one of the most demonstrated therapeutic effect of mesenchymal stem cells is mediated by the secretome, we tested if there were differences in the immunomodulating capability of the fractionated sub-populations of hAFSCs. The quantification of immunomodulating factors clearly indicated the conditioned medium secreted by final eluted cells as the richest in some of these components, namely, TGF-β and HGF-1. TGF-β is a potent anti-inflammatory cytokine that enhances the immunomodulatory properties of stem cells and, in line with other soluble factors associated with regenerative processes, HGF-1 possesses immune modulatory activity as well [46].

Interestingly, even if not in a significant manner, IL-6 seemed to be more concentrated in initial eluted cell secretome. IL-6 is a broad-acting cytokine involved in the control of the immune response as well as stem cell development and regulation. However, IL-6 has context-dependent pro- and anti-inflammatory properties. For example, IL-6 is considered a key driver of the pro-inflammatory IL-17-secreted by CD4+ or CD8+ T cells [47]. In light of this consideration, the IL-6 minor level in final eluted cell CM could be an advantage.

These data are consistent with the observation that viability of PBMC was more affected by final eluted cell CM, rather than in the presence of initial eluted cell secretome. Indeed, apoptosis was more intensely promoted by final eluted cell CM which contains a higher concentration of immunomodulating factors.

## 5. Conclusions

Amniotic fluid is a very interesting source of stem cells but gives rise to heterogeneous populations due to the different tissue origin, even if isolated for stem cell factor receptor c-kit. Here, we demonstrated that this new technology allows discriminating, according to physical parameters, the different cell types within a heterogeneous population kept in culture for different passages without any other additional manipulation. We obtained two sub-population of cells, showing different proliferation ability, senescence activation, stemness and differentiation proprieties.

## Figures and Tables

**Figure 1 cells-10-00158-f001:**
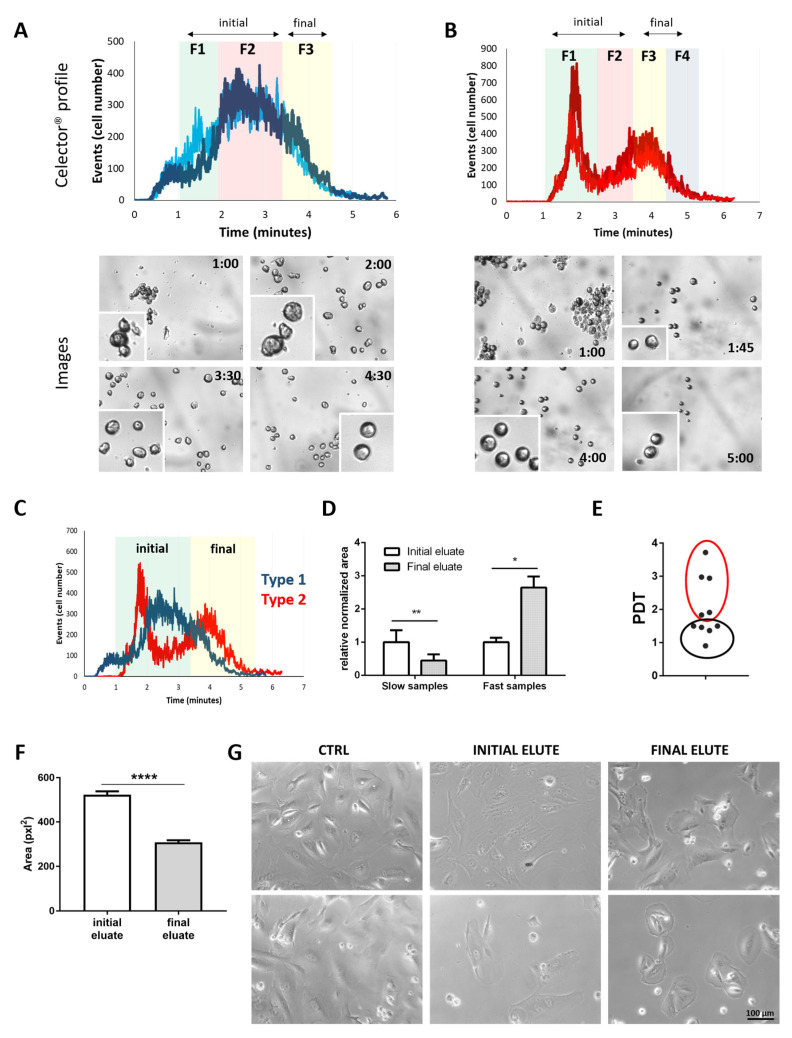
Celector^®^ sorting profiles of hAFSC samples. hAFSCs populations showed two types of profiles. Type (**A**) samples were divided into 3 fractions (F1, F2 and F3) while type (**B**) in 4 fractions (F1, F2, F3 and F4). Representative images and magnifications of eluting cells during time of analysis are shown below each profile. (**C**) The overlay of the two graphs are shown. In this graph is shown the division in initial and final eluate that was made to simplify all readouts. (**D**) The column graph shows the relative normalized area under each part of the Celector^®^ profile curve (Initial eluate = F1/F2; Final eluate = F3/F4). (**E**) The proliferative profile of hAFSCs populations is calculated by doubling population time (PDT) at passage three of the two groups of samples. (**F**) The area of eluting cells for each fraction in pixel^2^. It is shown how final eluted cells are smaller, while initial cells have a similar-fibroblastic morphology with larger and more elongated cytoplasm. (**G**) hAFSCs representative images of initial and final eluted cells after few days of culture post Celector^®^ selection. Statistical analysis: * *p* value < 0.05; ** *p* value < 0.01; **** *p* value < 0.0001.

**Figure 2 cells-10-00158-f002:**
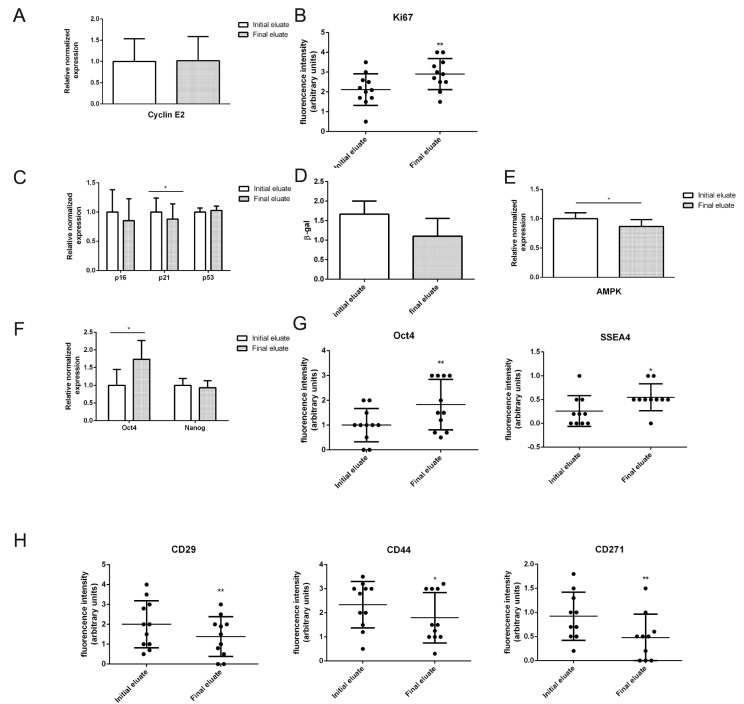
qPCR and immunofluorescence analysis of hAFSCs fractions. (**A**,**C**,**D**,**E**,**F**) Normalized expression of cyclin E2, p16, p21, p53, β-gal, AMPK, Oct4 and Nanog transcripts. Tubulin and GAPDH expression were used as housekeeping. Six different samples were analysed. (**B**,**G**,**H**) Distribution plots of fluorescence signal captured in confocal images (Appendix A) showing Ki67, Oct4, SSEA4, CD29, CD44, CD271. A paired test was applied. * *p* value < 0.05; ** *p* value < 0.01.

**Figure 3 cells-10-00158-f003:**
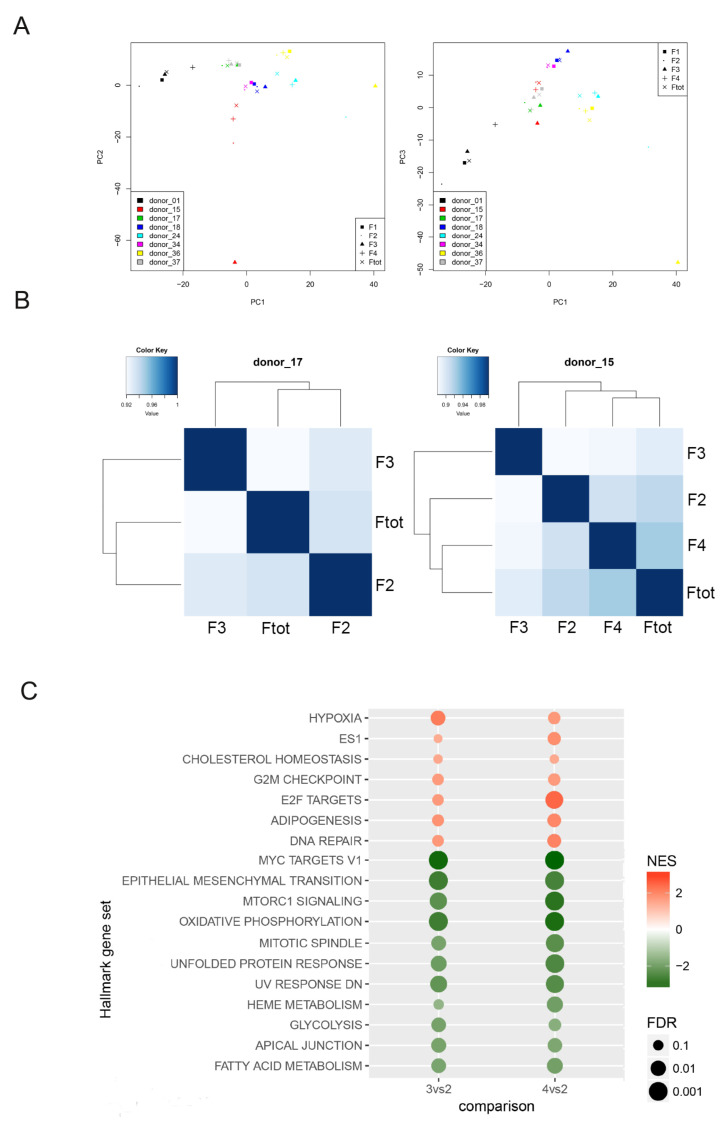
RNSseq analysis of hAFSCs fractions. (**A**) Principal component analysis reporting distribution of 8 samples along the principal component 1 (PC1), principal component 2 (PC2) and principal component 3 (PC3). The heterogeneity among patients and fractions can be noticed. (**B**) Two graphs of Pearson Correlation Analysis representative of samples sorted in 3 or 4 fractions are shown in order to highlight the heterogeneity among the fractions. (**C**) Pathways differently expressed with the same trend for both F3 versus F2 and F4 versus F2. 18 pathways differently expressed according to software Gene Set Enrichment Analysis (GSEA) analysis and hallmark dataset. In red pathways upregulated in F3 and F4, while in green pathways downregulated in these fractions compared to F2. The table represents the Normalized Enrichment score (NES) and the False discovery rate value (FDR). FDR < 0.25.

**Figure 4 cells-10-00158-f004:**
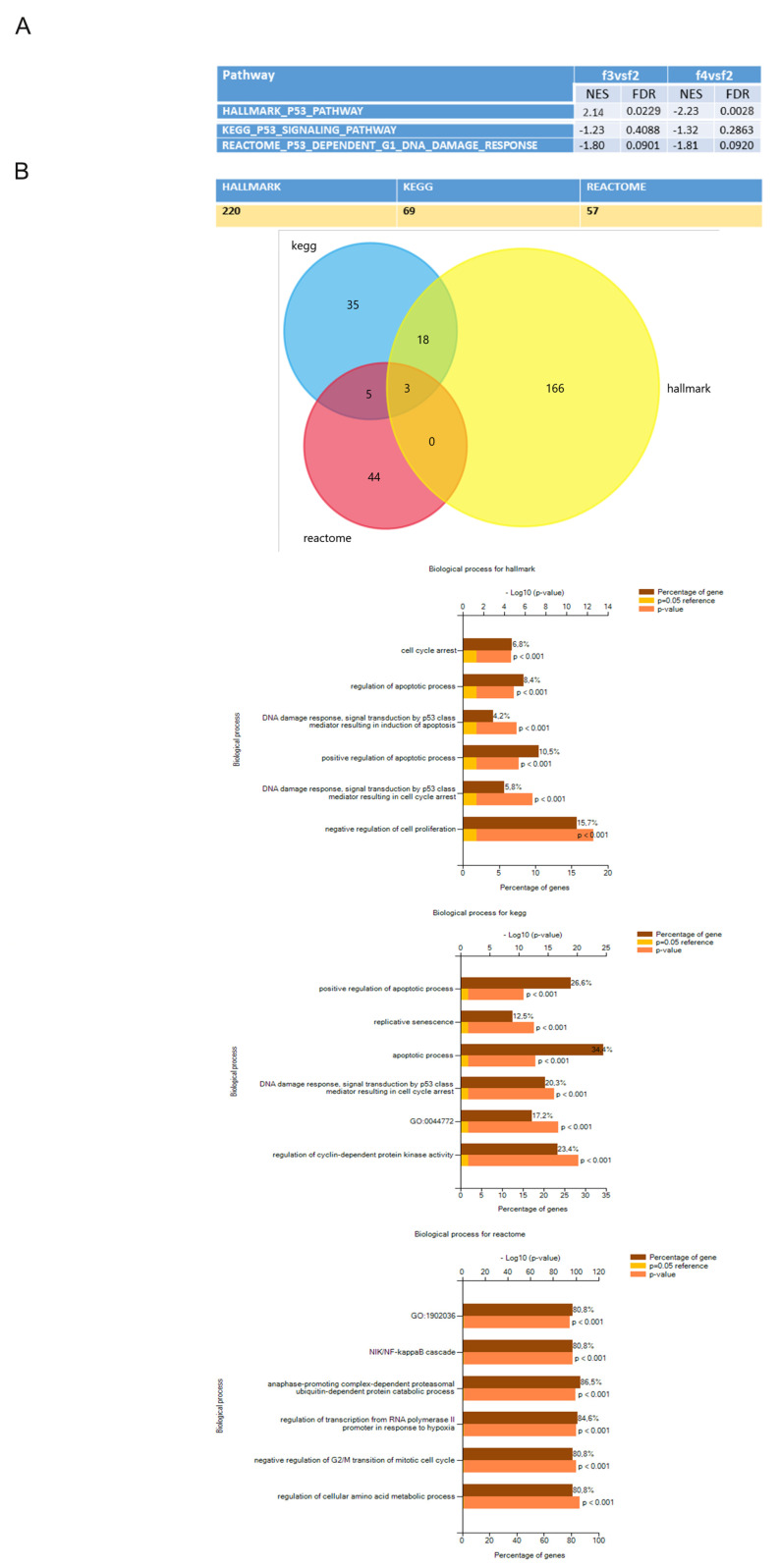
Evaluation of p53 pathway expression in different datasets. (**A**) Differential expression of p53 pathway in F3/F4 fractions versus F2 according to Hallmark, Kegg and Reactome. The table reports the Normalized Enrichment score (NES) and the False discovery rate value (FDR). (**B**) In the table above the number of genes of each dataset is reported. The bar graphs show at which biological processes genes composing each dataset can be ascribed. In particular, the brown bar shows the percentage of genes belonging to each biological process, the yellow bar shows the reference—log10 *p* value and the orange bar reports the—log10 *p* value of each process.

**Figure 5 cells-10-00158-f005:**
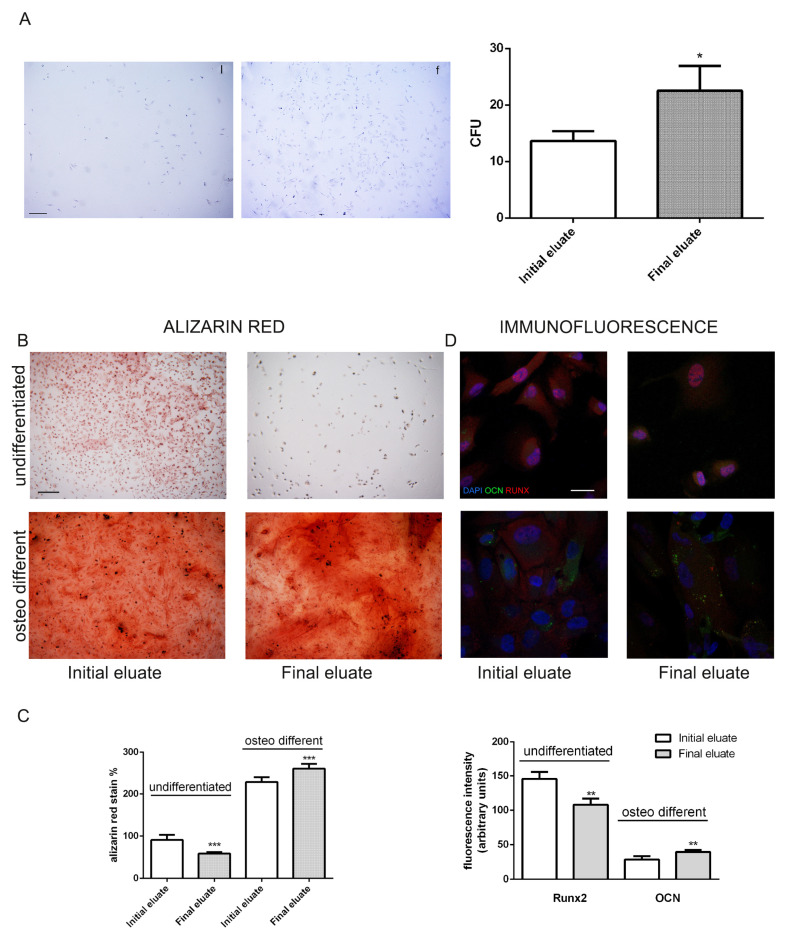
Evaluation of clonogenic and osteogenic differentiation potential of hAFSCs fractions. (**A**) hAFSCs derived from initial or final eluates were compared for proliferation capability. Representative images of colonies and the relative graph of the number of colony forming units (CFU) obtained. Bar = 100 µm. * *p* value < 0.05. (**B**) Differentiation capability of initial or final eluates into the osteo mesenchymal lineage. Osteogenic differentiation after three weeks of exposure to osteogenic medium. Evaluation of calcium deposition in the extracellular matrix through Alizarin Red staining: the intensity of red staining is related to the calcium presence typical of mineralizing tissue. Bar = 100 µm. (**C**) The graph shows the absorbance measured with spectrophotometer of the solubilized alizarin staining. *** *p* value < 0.001. (**D**) Representative confocal images of initial or final eluates of hAFSCs, exposed or not to osteogenic medium, labelled with DAPI (blue), anti-osteocalcin (OCN) in green and anti-Runx2 in red. Bar = 10 µm. The graph shows the fluorescence intensity of Runx2 in not differentiated samples and OCN for differentiating condition. ** *p* value < 0.01.

**Figure 6 cells-10-00158-f006:**
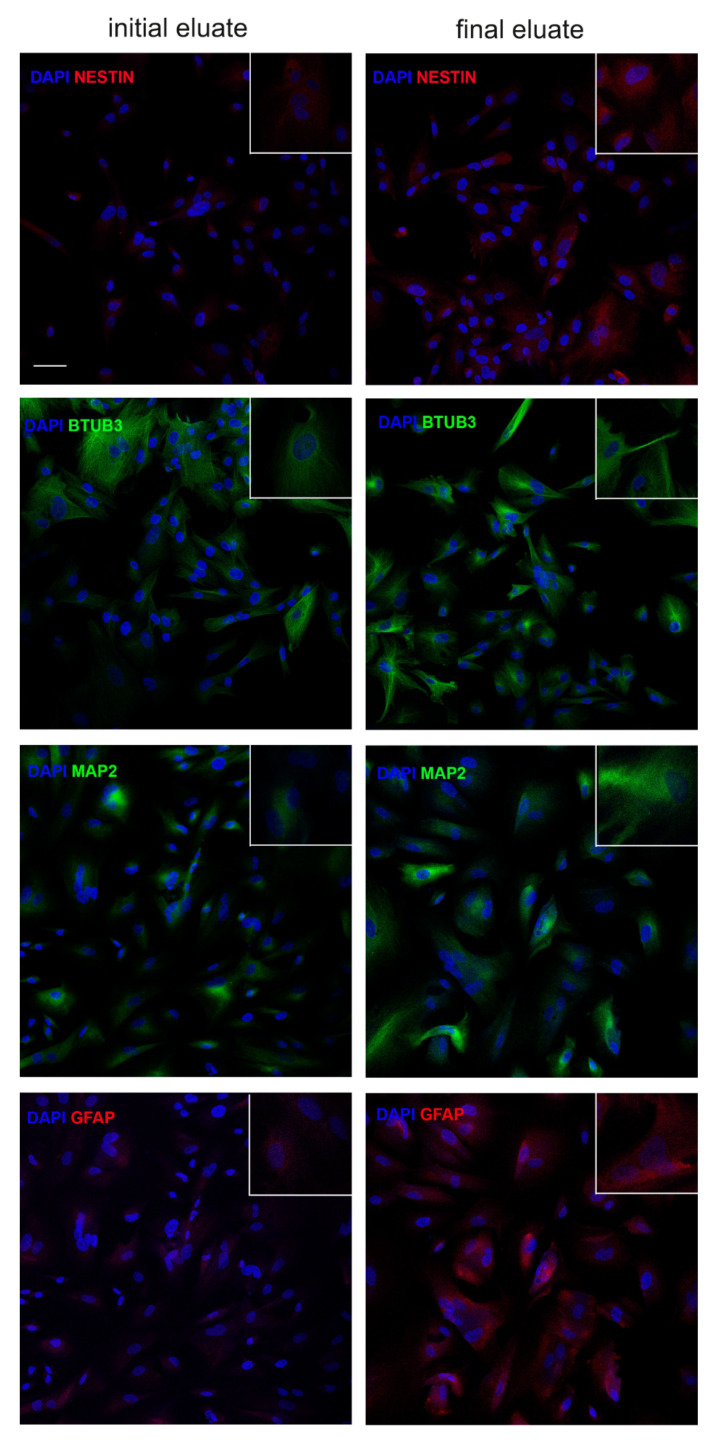
Evaluation of neurogenic differentiation potential of hAFSCs fractions. Differentiation capability of initial or final eluates into the neural lineage. Representative confocal images (and magnifications) of initial or final eluates of hAFSCs, exposed or not to neurogenic medium for 10 days, labelled with DAPI (blue), anti-nestin and anti-GFAP in red and anti- β3-Tubulin (BTUB3) and MAP2 in green. Bar = 20 µm.

**Figure 7 cells-10-00158-f007:**
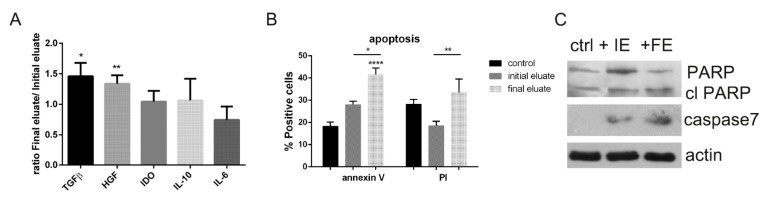
Analysis of the secretome produced by hAFSC fractions. (**A**) Graph representing ELISA tests for TGFβ, HGF, IDO, IL-10 and IL-6 analysed in condition medium (CM) produced by initial and final eluted cells obtained from 3 donors. The ratio between final and initial CM is shown. *T test* statistical analysis was performed to evaluate the difference between CM of final cells compared to the one of initial cells. * *p* value < 0.05; ** *p* value < 0.01. (**B**) Graph showing the percentage of PBMCs positive for Annexin V and PI after exposure for 4 days to CM produced by initial and final eluted cells. * *p* value < 0.05; ** *p* value < 0.01; **** *p* value < 0.0001. (**C**) Western blot analysis of lysates of PBMCs exposed to initial (IE) and final eluted (FE) hAFSCs revealed for PARP, caspase 7 and actin as loading control.

**Table 1 cells-10-00158-t001:** The list of primers used in the experiments.

Gene	Sequence	RefSeq Accession n
hNANOG	Fw cagtctggacactggctgaa Rv cacgtggtttccaaacaaga Probe 55	NM_024865
hPOU5F1 (OCT4)	Fw tgagtagtcccttcgcaagc Rv gagaaggcgaaatccgaag Probe 60	NM_002701.5
hCDKN2A (p16)	Fw gagcagcatggagccttc Rv cgtaactattcggtgcgttg Probe 67	NM_000077.4
hCDKN1A (p21)	Fw tcactgtcttgtacccttgtgc Rv ccgttttcgaccctgagag Probe 32	NM_000389.4
hTP53	Fw gctcaagactggcgctaaaa Rv gtcaccgtcgtggaaagc Probe 32	NM_001276760.1
AMPK	Fw tctcaggaggagagctatttgatt Rv gaacagacgccgactttcttt Probe 42	NM_006251.5
hCCNE2 (cyclin E2)	Fw ccccaagaagcccagataat Rv caggtggccaacaattcct Probe 35	NM_057749.2

## Data Availability

Bioinformatic data are available in GEO website (https://www.ncbi.nlm.nih.gov/geo/) with accession ID GSE164692.

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
