# Peer review of "Unravelling Heterogeneity of Amplified Human Amniotic Fluid Stem Cells Sub-Populations"

_cells, 2021, doi:10.3390/cells10010158_

Round 1

Reviewer 1 Report

In this study, the authors investigate the properties of human amniotic fluid stem cells (hAFSCs) using the Celector, a cell separation machine developed by StemSel, which is a new machine that uses chromatographic principles to separate cells without immunological labels. Using this machine, the authors success to analyze heterogeneous hAFSCs and separate them into several different cell populations. Overall, the paper is well-written, but I have some concerns.

1) The authors should also analyze the surface antigens of the isolated cell populations in each fraction by flowcytometry. The pattern of surface antigens in the cell populations isolated by the Celector is an important subject of study. If the surface antigens of all the cell populations show a similar pattern, then the usefulness of the Celector as claimed by the authors can be more strongly stated. If the expression patterns of the surface antigens were different, it might suggest a new isolation method for laboratories that cannot use the Celector.

2) The authors did not perform in vivo studies. In order to demonstrate the concept of using hAFSCs for regenerative medicine, it is necessary to study them in animal models. However, due to the limitation of time, I think it would be better if this point is considered in the future.

The manuscript as a whole is well written. When the authors present the additional data for question 1, I am willing to accept this manuscript for publication.

Author Response

Dear Editor,

We thank you and the reviewers for the opportunity to improve the paper. We answered to all the criticisms underlined by the reviewers. We hope that the manuscript is now suitable for publication in Cells.

Reviewer 1

We really thank the reviewer for the comments and suggestions to our study. Here below the answer to the comments.

1)         The authors should also analyze the surface antigens of the isolated cell populations in each fraction by flowcytometry. The pattern of surface antigens in the cell populations isolated by the Celector is an important subject of study. If the surface antigens of all the cell populations show a similar pattern, then the usefulness of the Celector as claimed by the authors can be more strongly stated. If the expression patterns of the surface antigens were different, it might suggest a new isolation method for laboratories that cannot use the Celector.

We agree with the reviewer that the presence of surface antigens is important, but this evaluation of mesenchymal markers, such as the CD29, CD44 and CD271, was already shown in the graphs of Figure 2 G representing the quantification of immunofluorescence signals obtained in each fraction. As reported in M&M section, the cell fluorescence signal was quantified (for 5 squares with 15 cells each) using ImageJ and applying the following formula:

Corrected Total Cell Fluorescence (CTCF) = Integrated Density – (Area of selected cell X Mean fluorescence of background readings).

This method for surface antigen quantification can be considered an alternative to cytofluorimetric analysis [ImageJ for microscopy, TJ Collins - Biotechniques, 2007] since it evaluates the intensity for each single cell, being not permeabilized in order to select the surface antigens only.

As previously reported, initial and final eluate cells positively expressed the aforementioned surface markers, even though differences were noted: the final eluate cells displayed a minor presence of all the mesenchymal markers, but the stemness markers, are more retained. Basing on this result, we state that Celector® can be applied to separate the cell population in order to have the “more mesenchymal part” or the “more stem part”.

2)         The authors did not perform in vivo studies. In order to demonstrate the concept of using hAFSCs for regenerative medicine, it is necessary to study them in animal models. However, due to the limitation of time, I think it would be better if this point is considered in the future.

We agree with the reviewer’s opinion. The in vivo experiments could be conducted in an inflammatory model, such as drug induced- osteoarthritis in rat knees, since we have recently demonstrated the efficacy of hAFSC-EVs in this pathology [Zavatti et al., 2020]. However, this new set of experiment will take other 6 months minimum and a new approval from the Bioethical Committee of the Italian National Institute of Health.

Reviewer 2 Report

In this study, Casciaro and co-Authors provide a comprehensive characterization of hAFSC subpopulations by means of label-free flow-based fractionation protocol. The study offers useful insights about hAFSC heterogeneity that may have relevant impact for their therapeutic exploitation within the regenerative medicine field. I think the manuscript could be improved by minor revisions with few additional experimental analyses. My suggestions are as following.

  • hAFSC heterogeneity has been reported by several independent groups; in 2012 Moschidou et al. demonstrated that hAFSC include both small and larger granular cells based on the expression of SSEA3 expression (Moschidou et al.Mol Ther 2012 Oct;20(10):1953-67). It would be interesting to assess whether the hAFSC subtractions isolated and characterized here by the Authors are related to SSEA3 expression as well;
  • reliable expression of the pluripotency marker OCT4 by hAFSC has been recently debated (Vlahova et al. Sci Rep 2019; 9(1):8126); indeed, the immunostaining representative pictures reported in Supplementary Figure show a weak nuclear signal of such marker. Authors shall consider considering Nanog as more reliable pluripotency marker by providing immunostaining analyses and corresponding pictures;
  • Authors shall consider including smooth muscle lineage commitment within the differentiation potential evaluated for the hAFS subtractions; indeed, functional acquisition of smooth muscle phenotype by hAFSC has been reported to be relevant for tissue engineering applications (Ghionzoli et al. FASEB J. 2013 Dec;27(12):4853-65);
  • Authors may want to include also senescence-activated beta-galactosidase staining to exclude activation of senescence within the sorted hAFSC subtraction, since cells in the final elute in Figure 1G show a quite enlarged morphology;
  • It is not very clear to me why Authors sorted hAFSC subfractions at passage 5-6 and not soon after CD117 selection, when cells reached 70% confluence after the immunomagnetic separation;
  • Authors should provide more details about the isolation of hAFSC-conditioned media (i.e. how long hAFSC have been cultured in serum free condition for before collecting the cell-conditioned medium; the yield of the cell-conditioned media from the sorted hAFSC subfractions in terms of protein concentration/million of secreting cells; whether any difference in the yield of the subfractions has been detected, etc).

Author Response

Dear Editor,

We thank you and the reviewers for the opportunity to improve the paper. We answered to all the criticisms underlined by the reviewers. We hope that the manuscript is now suitable for publication in Cells.

Reviewer 2

We really thank the reviewer for the suggestions and encouraging comments to our study. Here below the answer to the comments.

  1. hAFSC heterogeneity has been reported by several independent groups; in 2012 Moschidou et al. demonstrated that hAFSC include both small and larger granular cells based on the expression of SSEA3 expression (Moschidou et al.Mol Ther 2012 Oct;20(10):1953-67). It would be interesting to assess whether the hAFSC subtractions isolated and characterized here by the Authors are related to SSEA3 expression as well.

We agree with the reviewer that the expression of SSEA3 in each fraction could be interesting to analyze, but, since the editor gave us only 5 days and we do not have the antibody, we cannot investigate this point. However, c-kit positive cells, isolated by De Coppi in Nature Biotechnology 2007 and more recently by Qin et al Int J Biol Sci 2016, were positive for stage-specific embryonic antigen (SSEA)-4 but did not express other surface markers characteristic of ES12,13 and embryonic germ (EG) cells14, SSEA-3 and Tra-1-81.

2          reliable expression of the pluripotency marker OCT4 by hAFSC has been recently debated (Vlahova et al. Sci Rep 2019; 9(1):8126); indeed, the immunostaining representative pictures reported in Supplementary Figure show a weak nuclear signal of such marker. Authors shall consider considering Nanog as more reliable pluripotency marker by providing immunostaining analyses and corresponding pictures.

Actually, we have already shown in Figure 2 E that the RT-PCR analysis of Nanog reveals similar, but weak levels of this marker in both the eluates, therefore we decided to not evaluate this protein by IF. Conversely, Oct4 was differently expressed, considering RT-PCR experiments, so we confirmed this trend by IF analysis that allowed to discriminate between the nuclear signal and the cytosolic one. In our opinion, the nuclear signal in the final eluate samples is definitely clear and higher, compared to the one of initial eluate, sustaining the RT-PCR data. Oure findings are consistent with results reported in a recent paper (Investigating the expression of pluripotency-related genes in human amniotic fluid cells: A semi-quantitative comparison between different subpopulations, from primary to cultured amniocytes. Hoseini SM, et al. Reprod Biol. 2020 Sep;20(3):338-347) where the authors demonstrated that there is non-significant difference in gene’s expression between epithelioid and fibroblastoid lines (except for OCT4 that was higher in epithelioid subpopulation).

3          Authors shall consider including smooth muscle lineage commitment within the differentiation potential evaluated for the hAFS subfractions; indeed, functional acquisition of smooth muscle phenotype by hAFSC has been reported to be relevant for tissue engineering applications (Ghionzoli et al. FASEB J. 2013 Dec;27(12):4853-65);

We agree with the reviewer’s comment on the relevance of smooth muscle differentiation potential of hAFSC, however, due to the lack of time, we cannot perform this experiment. We decided to evaluate the differentiation potential into the ectodermal lineage (neurogenic differentiation), in order to better sustain the higher plasticity of final eluate, instead of a differentiation into another mesoderm- derived cytotype. This assay was performed under the suggestion of the previous reviewers. Moreover, we would like to specify that this is a second submission, derived by the suggestion of the Editor Billie Jiao in August 2020 to perform, in few months, additional experiments only on the immunomodulatory effect of fractionated AFSC.

4)         Authors may want to include also senescence-activated beta-galactosidase staining to exclude activation of senescence within the sorted hAFSC subtraction, since cells in the final elute in Figure 1G show a quite enlarged morphology;

Following the reviewer’s suggestion, we added beta-galactosidase evaluation to Figure 2. Here we report the obtained data (from 3 different AFSC samples) that are consistent with the p21 result, even if the p value is only 0.37:

5)         It is not very clear to me why Authors sorted hAFSC subfractions at passage 5-6 and not soon after CD117 selection, when cells reached 70% confluence after the immunomagnetic separation;

In order to study sorted cells by Celector®, a high number of starting cells is requested, at least around 2 million of cells, to obtain a sufficient material for downstream analysis. The reviewer has to take in consideration the necessity to amplify AFSCs before performing cKit selection and a following expansion for the Celector® selection because cKit+-AFSCs are around 1-5% of the entire amniotic fluid cell populations. This amplification should be done even in the case of cell therapy or secretome application, therefore the number of AFSC that we used for the Celector® separation are similar to the one that could be used in clinical practice.

6)         Authors should provide more details about the isolation of hAFSC-conditioned media (i.e. how long hAFSC have been cultured in serum free condition for before collecting the cell-conditioned medium; the yield of the cell-conditioned media from the sorted hAFSC subfractions in terms of protein concentration/million of secreting cells; whether any difference in the yield of the subfractions has been detected, etc).

Here we provide additional details on the experimental procedure performed for conditioned medium collection. Depending on the sample, a different number of cells was collected from initial o final eluate fractions, since every sample is characterized by a different cell composition. However, cells were seeded at the same density and AFSC subfractions were cultured without serum for 4 days. One mL of conditioned medium was derived from 50000 cells. Similar protein concentration was found in all CM, analyzing by Bradford test. These details were added to M&M section.

Reviewer 3 Report

The article by Casciaro et al et submitted to Cells and titled “Unraveling Heterogeneity of Human Amniotic Fluid Stem Cells Sub‐Populations” describes the identification of intra‐population differences that can influence the stemness profile of hAFSC taking advantage of a new device.  

The article is a hot topic and is of interest for this journal and last but not least very well written, despite that some major revisions must be assessed.

Results: 

Paragraph

- Authors report that they perform the selection using the Celector at passage 5-6 starting from c-kit + cells. Was the positivity for c-kit also evaluated immediately before the isolation process or the cells have been described as c-lit positive because initially isolated by this strategy?

Furthermore, the variability in the different fractions identified by the instrument is characteristic of normal heterogeneity of stem cells kept in culture for different passages. Indeed, some cells undergo senescence as also indicated by the same authors describing the cells as “Enlarged morphology and reduced cell cycle progression are considered hallmarks of cellular senescence. Major regulators of cellular senescence are the cyclin‐dependent kinase inhibitors 328 p16INK4A and p21WAF1 [30]. Interestingly, we observed a significant increase of p21 expression in 329 the initial eluate cells (Figure 2B)” 

Could the authors make a comparison even with cells derived from the first passages (passage1-3) and see report the differences observed? Otherwise, authors must soften the title and modify the text

-Authors reported differences in the positivity for the ki67 marker, however, authors should perform a synchronization of the cells with nocodazole or other drugs before to perform the isolation fo the different fractions to correctly compare the different proliferative potential

-Authors should improve the resolution of the figures. Overall figure 4, when enlarging the text become crumbled

Authors isolated and kept in culture the PBMC for twenty in RPMI complete medium, however, in general, the PBMC must be used early or at least in the few hours upon isolation otherwise monocytes adhere and start to activate and in general is not completely representative of the correct response of the immune cells isolated from the peripheral blood. Furthermore, the authors reported increased apoptosis of the PBMC. Looking with a glance at the regenerative medicine translation of hAFSC, increased apoptosis can negatively or partially interfere with the immune. Indeed, the creation of apoptotic bodies can, in part, prompt the macrophages to skew from the M1 inflammatory subset towards the anti-inflammatory M2. On the other hand, other immune cells can instead be activated by the release of inflammatory cytokines and DAMP from the apoptotic cells. How do the authors explain this process?

- In the conclusion section, line 558 authors stated “due to the different tissue origin, even if isolated for stem cell factor receptor c‐kit”. Authors ensure that the differences observed are due to the heterogeneity of the source and not rather to an inherent heterogeneity of the population kept in culture. Did they perform any staining or perform a gene expression analysis that allows them to discriminate among the different sources of origin or epigenetic profiling? Currently, authors can conclude that the Celector device allows discriminating, according to physical parameters, the different cell types within a heterogeneous population kept in culture for different passages. Please modify toning down their conclusion

Author Response

Dear Editor,

We thank you and the reviewers for the opportunity to improve the paper. We answered to all the criticisms underlined by the reviewers. We hope that the manuscript is now suitable for publication in Cells.

Reviewer 3

We really thank the reviewer for the suggestions and encouraging comments to our study. Here below the answer to the comments.

  • Authors report that they perform the selection using the Celector at passage 5-6 starting from c-kit + cells. Was the positivity for c-kit also evaluated immediately before the isolation process or the cells have been described as c-lit positive because initially isolated by this strategy?

Following the reviewer’s suggestion, we performed IF analysis of c-kit positivity in each fraction, compared to the unfractionated sample at the time of Celector® selection. The analysis performed in a sample of hAFSC revealed that unfractionated sample displayed a weak positivity in all cells, and that the initial and the final eluate showed a similar expression’s signal. Since this result was obtained with only one sample and that it does not suggest a difference between fractions, we did not add the data to the manuscript.

  • Furthermore, the variability in the different fractions identified by the instrument is characteristic of normal heterogeneity of stem cells kept in culture for different passages. Indeed, some cells undergo senescence as also indicated by the same authors describing the cells as “Enlarged morphology and reduced cell cycle progression are considered hallmarks of cellular senescence. Major regulators of cellular senescence are the cyclin‐dependent kinase inhibitors 328 p16INK4A and p21WAF1 [30]. Interestingly, we observed a significant increase of p21 expression in 329 the initial eluate cells (Figure 2B)” Could the authors make a comparison even with cells derived from the first passages (passage1-3) and see report the differences observed? Otherwise, authors must soften the title and modify the text.

In order to study sorted cells by Celector®, a high number of starting cells is requested, at least around 2 million of cells, to obtain a sufficient material for downstream analysis. The reviewer has to take in consideration the necessity to amplify AFSCs before performing cKit selection and a following expansion for the Celector® selection because cKit+-AFSCs are around 1-5% of the entire amniotic fluid cell populations. This amplification should be done even in the case of cell therapy or secretome application, therefore the number of AFSC that we used for the Celector® separation are similar to the one that could be used in clinical practice.

Following the reviewer’s suggestion, we modified the text and the title, specifying that these are long-term cultured AFSC.

  • Authors reported differences in the positivity for the ki67 marker, however, authors should perform a synchronization of the cells with nocodazole or other drugs before to perform the isolation fo the different fractions to correctly compare the different proliferative potential

We did not perform a cell synchronization. However, the RNAseq analysis confirmed that the final eluate is enriched in the G2M checkpoint pathway, in support of our initial hypothesis that the proliferative potential of this fraction was significantly improved.

  • Authors should improve the resolution of the figures. Overall figure 4, when enlarging the text become crumbled.

Following the reviewer’s suggestion, we improved the resolution of figure 4.

  • Authors isolated and kept in culture the PBMC for twenty in RPMI complete medium, however, in general, the PBMC must be used early or at least in the few hours upon isolation otherwise monocytes adhere and start to activate and in general is not completely representative of the correct response of the immune cells isolated from the peripheral blood. Furthermore, the authors reported increased apoptosis of the PBMC. Looking with a glance at the regenerative medicine translation of hAFSC, increased apoptosis can negatively or partially interfere with the immune. Indeed, the creation of apoptotic bodies can, in part, prompt the macrophages to skew from the M1 inflammatory subset towards the anti-inflammatory M2. On the other hand, other immune cells can instead be activated by the release of inflammatory cytokines and DAMP from the apoptotic cells. How do the authors explain this process?

We agree with the reviewer that the observation of induced apoptosis by conditioned media (CM) can be negatively interpreted. However, we previously demonstrated (Beretti et al. Biofactors 2018) that the extracellular vesicles included into the AFSC-CM can display a finer immunomodulating effect than the separately CM. The immunomodulatory effect of the total secretome produced by AFSC, divided in exosomes and the residual secretome part, was evaluated on the cell cycle of PBMC by flow cytometry. We presented the evidence that the secretome of AFSC-modulates PBMC cell cycle, while the one present in amniotic fluid has not significant effect. In particular the induction of apoptosis, which we previously observed in 2015 with the total secretome treatment, was stimulated only by fraction of AFSC deprived of exosomes. Moreover, to understand the immunomodulatory potential of AFSC secretome, we analyzed the modifications in percentage of cells positive for the T-lymphocytes markers, such as CD4 for the helper and CD8 for the cytotoxic population. We focused our attention on these types of T- lymphocytes since it has been recently demonstrated that hAFSC vesicles reduced the maturation of memory B cells (in PWM- stimulated PBMC), but did not influence the polarization of T reg [Balbi et al., 2017]. We found that lymphocyte T helper seems to be the subpopulation most affected in the proliferation decrease. Furthermore, we noticed that exosome deprived AFSC fraction also reduces the percentage of cells in S and G2/M phases. On the other hand, the exposure to AFSC-derived exosomes decreases only, but more intensely, the lymphocyte’s proliferation, as demonstrated also by BrdU experiments. These data support the hypothesis that the entire secretome of stem cells differently affects immune response.

Since this is a second submission, derived by the suggestion of the Editor Billie Jiao in August 2020 to perform in few months additional experiments only on the immunomodulating effect of fractionated AFSC (even by ELISA tests for HGF, TGFbeta, IL-6 and IL-10), we decided to follow our first demonstrated observations on AFSC secretome: conditioned medium (CM) of hAFSCs induces apoptosis in PBMCs [Maraldi et al., 2015], trying to dissect the effect of secretome of the two fractions, initial and final eluate. Therefore, we arranged an experimental design similar to the one previously tested: in brief, CM were collected after 4 days in culture (deprived of serum). Then, PMBCs were exposed to CM for other 4 days. We did not observe any cell adhesion in the 24 hours in which PBMC were in culture before the CM treatment.

Data on extracellular vesicles obtained from CM of fractionated AFSC could be part of a new study that needs much more time.

  • In the conclusion section, line 558 authors stated “due to the different tissue origin, even if isolated for stem cell factor receptor c‐kit”. Authors ensure that the differences observed are due to the heterogeneity of the source and not rather to an inherent heterogeneity of the population kept in culture. Did they perform any staining or perform a gene expression analysis that allows them to discriminate among the different sources of origin or epigenetic profiling? Currently, authors can conclude that the Celector device allows discriminating, according to physical parameters, the different cell types within a heterogeneous population kept in culture for different passages. Please modify toning down their conclusion.

We agree with the reviewer’s comment, so we modified the conclusion.

This manuscript is a resubmission of an earlier submission. The following is a list of the peer review reports and author responses from that submission.

Round 1

Reviewer 1 Report

This paper describes the characteristics of the various fractions of hAFSCs isolated using the Celector®, suggesting a proof-of-concept of the importance to select certain cellular fractions with the highest potential to use in regenerative medicine.

Overall, the authors have made a good attempt at adding value to profile hAFSCs. However, the impact is lost by a limited investigation of biological heterogeneity. In my opinion, a major revision of manuscript is needed before it can be accepted for publication.

Major point

1) To clarify the level of biological heterogeneity, cytokines and growth factor secretions in each fraction of hAFSCs should be compared. Moreover, changes in therapeutic effect due to in each fraction of hAFSCs should be demonstrated both in vitro and in vivo.

2) As far as I know, until now, two different kind of hAFSCs have been reported (CD117 selection and morphology during culture). Among them, the author used CD117 positive amniotic fluid cells as hAFSCs in this study. As described by De Coppi et al. , it has been reported that the CD117 method is performed using amniotic fluid cells before cell culture. However, in this study, CD117 immunoselection was performed using amniotic fluid cells after cell culture (line 107 and ref.20). I think that the cell population obtained by these two methods may be different.

Therefore, the author should describe the method used to isolate CD117 amniotic fluid cells from amniotic fluid cells in detail.

Reviewer 2 Report

The authors report on an experimental study where they investigated the heterogeneity of human amniotic fluid stem cell (AFSC) sub-populations that were separated by a new technology.

This study addresses a very important topic in stem cell biology, i.e. cell fraction heterogeneity, which represents a possible limitation for procedure standardization. The manuscript is well structured, but would benefit from a review by a native English speaker as some minor language mistakes and typos are found throughout the text and in the figures.  

To prove that the different fractions still have the reported differentiation potential towards cells of all three germ layers, the authors should test the ability of the different fractions to differentiate in more cell types (beyond the osteogenic lineage).

The authors report that two subpopulations of c-kit+ cells have different proliferation ability, senescence activation, stemness and differentiation proprieties. These assessments are indeed interesting. However, the study would benefit from a validation of these prosperities by testing the functionality of the different fractions on a given platform. As there is ample literature on the use of AFSCs as a treatment for several conditions, the authors could choose an in vitro model to quickly assess the potential functional differences of the isolated fractions.

Reviewer 3 Report

The manuscript “Unravelling heterogeneity of human amniotic fluids cells sub-populations” describes about stemness profile of human amniotic fluid stem cells (hAFSCs). The authors compared the various cellular fractions of hAFSCs which could be very useful in the field of regenerative medicine. The authors used Celector technology to isolate the cellular fractions of hAFSCs based on physical properties. Overall, manuscript is very well written and informative to the scientific community. However, I have few concerns:

  1. Figure 1A: It is difficult to understand the size of cells from images included in figure 1A. is it possible to include high magnification images to see visual differences among various cellular fractions.
  2. The authors showed that final elutes are more proliferative in compare to initial elutes. I am curious to know about number of passages these cells can grow. The authors mentioned about p16 in line 305 but it not clear what authors want to say here. I would suggest to discuss about how many passages these cells can culture and how many cells can be generated from given number of cells.
  3. Figure 2 legend is not very clear. What is Figure 2C, 2D, 2F, and 2G? Please include the information in the figure legend. I would be nice to include one representative images for each marker along with intensity bar graph.
  4. Figure 5: please include the representative CFU-F images in figure 5. Scale bar is missing for alizarin red staining in figure 5B. spelling correction in figure 5C, “oteo different” should be written as osteo different or osteogenic differentiation.

Round 2

Reviewer 1 Report

In my opinion, to clarify the level of biological heterogeneity, cytokines and growth factor secretions in each fraction of hAFSCs must be compared. Moreover, changes in therapeutic effect due to in each fraction of hAFSCs should be demonstrated, at least in vitro. 

Reviewer 2 Report

I commend the authors for resubmitting the manuscript and for producing preliminary data that can be used towards my comments. I completely agree that given the tight deadline of 10 days, it is not feasible to perform the experiments outlined. I therefore ask the editor to extend the time required for this re-submission so that the authors can have ample time to complete the experiments for comment 2. 

Reviewer 3 Report

Authors answered all my queries and I do not have any further comments on the manuscript.